Spatial variation in coral reef fish and benthic communities in the central Saudi Arabian Red Sea

Khalil Maha T. maha.khalil@kaust.edu.sa 1
Bouwmeester Jessica 1 2
Berumen Michael L. 1
1 Red Sea Research Center, Division of Biological and Environmental Science and Engineering, King Abdullah University of Science and Technology , Thuwal , Saudi Arabia
2 Department of Biological and Environmental Sciences, College of Arts and Sciences, Qatar University , Doha , Qatar
Thompson Fabiano
Electronic publication date: 2017 Jun 6
Publication date: 2017
Volume: 5
Electronic Location ID: e3410
Received 2016 Aug 3; Accepted 2017 May 12
Copyright: ©2017 Khalil et al.
Copyright year: 2017
Copyright holder: Khalil et al.
License: This is an open access article distributed under the terms of the Creative Commons Attribution License, which permits unrestricted use, distribution, reproduction and adaptation in any medium and for any purpose provided that it is properly attributed. For attribution, the original author(s), title, publication source (PeerJ) and either DOI or URL of the article must be cited.
License URL: https://creativecommons.org/licenses/by/4.0/

Keywords: Inshore-offshore gradients, Coral cover, Diversity, Fish biomass, Saudi Arabia, Red Sea, Trophic structure, Community assemblages

Funding: King Abdullah University of Science and Technology URF/1/1389-01-01 Red Sea Research Center URF/1/1973-01-01 This study was funded by the King Abdullah University of Science and Technology through a Competitive Research Grant (URF/1/1389-01-01), the Red Sea Research Center (URF/1/1973-01-01), and baseline research funds to Michael L. Berumen. The funders had no role in study design, data collection and analysis, decision to publish, or preparation of the manuscript.

==============================
Local-scale ecological information is critical as a sound basis for spatial management and conservation and as support for ongoing research in relatively unstudied areas. We conducted visual surveys of fish and benthic communities on nine reefs (3–24 km from shore) in the Thuwal area of the central Saudi Arabian Red Sea. Fish biomass increased with increasing distance from shore, but was generally low compared to reefs experiencing minimal human influence around the world. All reefs had a herbivore-dominated trophic structure and few top predators, such as sharks, jacks, or large groupers. Coral cover was considerably lower on inshore reefs, likely due to a 2010 bleaching event. Community analyses showed inshore reefs to be characterized by turf algae, slower-growing corals, lower herbivore diversity, and highly abundant turf-farming damselfishes. Offshore reefs had more planktivorous fishes, a more diverse herbivore assemblage, and faster-growing corals. All reefs appear to be impacted by overfishing, and inshore reefs seem more vulnerable to thermal bleaching. The study provides a description of the spatial variation in biomass and community structure in the central Saudi Arabian Red Sea and provides a basis for spatial prioritization and subsequent marine protected area design in Thuwal.

Introduction

Despite the uniqueness of its environment and the fact that it possesses one of the longest coral reef systems in the world, coral reef ecology remains relatively understudied in the Red Sea in comparison to other biogeographical regions (Berumen et al., 2013). Detailed information on spatial patterns of fish biomass, fish densities, and structure of benthic and fish assemblages are available only for some parts of the Red Sea, primarily the Gulf of Aqaba and parts of Egypt (e.g., Bouchon-Navaro & Bouchon, 1989; Alwany & Stachowitsch, 2007).

Saudi Arabia has the largest stretch of Red Sea coastline (approximately 1,700 km) and is home to a variety of coral reef habitat types (e.g., Sheppard, Price & Roberts, 1992), yet there are relatively few accessible publications available from this region that report basic and detailed ecological patterns. Ecological information from the Saudi Arabian Red Sea is mostly confined either to reports prepared by collaborating regional and international organizations and published in grey literature (e.g., PERSGA/GEF, 2003) or to large-scale studies focused on regional trends and patterns (e.g., Roberts, Alexander & Ormond, 1992; Price et al., 1998; DeVantier et al., 2000; Roberts et al., 2016a, Roberts et al., 2016b). With the exception of a few recent studies (e.g., Furby, Bouwmeester & Berumen, 2013), little work has been done to characterize reef communities on small, local scales, which are appropriate for informing local resource-managers and decision makers (Margules & Pressey, 2000), and there are even fewer studies using detailed taxonomic resolution (e.g., fish or benthic species).

However, recent expansion of research activity in Saudi Arabia (Mervis, 2009) has begun to address questions about the functioning of Red Sea reefs at local scales (e.g., Davis et al., 2011; Jessen et al., 2013; Van der Merwe et al., 2014). One example is the thermal bleaching event that occurred in summer 2010 (Furby, Bouwmeester & Berumen, 2013, Pineda et al., 2013), which raised questions about the potential local impact of overfishing and coastal development on the inherent ability of reefs to recover from such major disturbances (resilience), particularly in the presence of climate change (Khalil, Cochran & Berumen, 2013). Ongoing research efforts and eventual conservation planning increasingly highlight the need for detailed assessments of local and regional (e.g., Roberts et al., 2016a; Roberts et al., 2016b) reef communities.

This study aimed to describe the reef communities off the coast of Thuwal in the central Saudi Arabian Red Sea by exploring spatial patterns of the biomass, density, and diversity of reef fishes at two different depths, with focus on important trophic and commercial groups. We also describe the cross-shelf and vertical spatial variation in benthic cover and in fish and benthic assemblages. We expected to find a cross-shelf gradient of increasing overall fish biomass and diversity with distance from shore due to typical environmental gradients in reef topography, bathymetry, sedimentation, food availability, or human impact, which are recurring patterns found in previously conducted cross-shelf analyses around the world (e.g., Fabricius, 2005; Aguilar-Perera & Appeldoorn, 2008; Nemeth & Appeldoorn, 2009; Malcolm, Jordan & Smith, 2010). We also expected to find clear spatial variation in fish species richness and assemblage co-occurring with any differences in benthic assemblage (Roberts & Ormond, 1987; Chabanet et al., 1997; Chong-Seng et al., 2012). Finally, we suggest potential explanations and implications of some of these spatial patterns based on comparisons to other parts of the world. The ultimate aim of the study was to provide a scientific basis for subsequent spatial prioritization and conservation planning (see Khalil, 2015) by highlighting local areas of high and low diversity or biomass.

Methods

Study site

The study area includes 355 patch reefs of varying sizes distributed within an area of about 2,200 km2 along approximately 70 km of the central Saudi Arabian coast. The furthest reef is about 25 km from shore. The coastline in this area is moderately developed, with two relatively large coastal towns and one small fishing town called Thuwal (22.28°N, 39.10°E) (Fig. 1). The area suffered from a severe bleaching event in the boreal summer of 2010, which had the highest impact on reefs closest to shore. Inshore reefs lost most of their adult coral cover up to a depth of 10 m and experienced a change in coral assemblage (Furby, Bouwmeester & Berumen, 2013).

Figure 1 Map of study area.

The study area and locations of surveyed reefs in the central Saudi Arabian Red Sea. Depth is color-coded as noted in the inset key. The black color represents the shallowest portion of reef areas (seasonally intertidal). Reef name abbreviations (see main text) are shown next to their respective, color-coded marker circles: red, inshore reefs; green, midshelf reefs; blue, offshore reefs. GPS coordinates are indicated by a decimal-degree grid on the left and top margins. Right-hand panels show the georgraphic location of the Red Sea and the locaition of the study site along its coast. (Map created in ArcMap, version 10.1, by MTK using various mapping sources freely available through ESRI®).

We surveyed nine reefs at various distances from shore (Fig. 1). The three offshore reefs (furthest from shore and adjacent to waters deeper than 200 m) were, from north to south, Abu Romah Reef (RR), Nazar Reef (NR), and Abu Madafi Reef (AMR). Midshelf reefs (closer to shore and adjacent to waters that are 50–200 m deep) were Al-Fahal Reef (FR), Al-Taweel Reef (TWR), and Abu-Henshan Reef (AHR). Inshore reefs (closest to shore and surrounded by waters around 20 m deep) were Abu Shosha Reef (ASR), Tahla Reef (TR), and East Fsar Reef (EFR). Typical of the region, these reefs are arranged in small clusters, with relatively large elongated reef patches oriented on a north-south axis and surrounded by smaller, rounder, patches and pinnacles. All study reefs have relatively steep walls dropping down to 20 m or deeper and very shallow reef tops, with the exception of inshore reefs which drop to a sloping seabed at 10–15 m (Sheppard, Price & Roberts, 1992).

Fish and benthic surveys

Surveys were conducted in May 2013 at two depths (on the reef wall at 10 m and on the reef crest at 1–3 m) at each of the nine reefs. On the inshore reefs, the deep transects at 10 m were at the bottom where the reef walls sloped to meet the sandy sea floor. All transects were located on the west sides of the reefs, exposed to prevailing winds, currents, and waves. Fish surveys were conducted along three belt transects at each depth (a total of six transects per reef), where a diver swam along the transects twice, first to record larger vagile fish (>18 cm TL) in 25 × 8 m belts and a second time to record smaller fish (<18 cm TL) in 25 × 4 m belts (following Sandin et al., 2008). Individual fishes were counted and their sizes were estimated and placed in categories of total length in cm (0–3, 4–5, 6–10, 11–15, 16–20…61–70, 71–80…101–150, 151–200 cm). Categories larger than 100 cm were merged as only two species (the moray eel Gymnothorax javanicus and the white tip reef shark Triaenodon obesus) were observed in these categories, and we were less confident in the accuracy of these size estimates. We did not attempt to count cryptic species (see Table S1 for a list of species observed) as these are poorly described in the Red Sea and require specific sampling methods (e.g., Tornabene et al., 2013).

Benthic surveys to determine live scleractinian (hard) coral cover, coral genus richness, and other benthic categories were conducted on the same transects as the fish surveys using the line-intercept method. Apart from hard coral genera, we recorded the cover of soft corals and zoanthids (to genus level when possible), sponges, crustose coralline algae (CCA), turf algae, and “other” algae. Transects for benthic surveys were 10 m long and located in the middle of each of the 25 m transects used for counting fish, making a total of six transects per reef, three at each depth. The transect length was chosen for its convenience in the field, to be comparable to previous studies done in this region (Furby, Bouwmeester & Berumen, 2013), and because it has been previously shown to be adequate for quantitative studies of coral cover (Beenaerts & Berghe, 2005). In order to minimize the impact of observer bias, all data were collected by the same divers (JB benthos, MLB fishes).

Biomass, abundance, and diversity estimations

Fish biomass and trophic composition

Fish biomass was calculated following Friedlander & DeMartini (2002) using the equation: W = a × Lb, where W is the weight of the fish in grams, L is its total length (TL) in cm and a and b are species-specific constants obtained from FishBase (Froese & Pauly, 2014) (see Table S1 for a list of species-specific constants). For the L value, we used the mid-range value of each TL size category. When several values of a and b were present in the database for a given species, we used an average of the available values, and when values were missing from the database, we used those provided for sister species, the genus, or the family. The average biomass of all species was then calculated in kg/100 m2 for each reef, and, from these values, we summarized the biomass of four trophic guilds (top predators, carnivores, herbivores, and planktivores) and three major groups of commercially targeted fish, which included 25 species in five subfamilies: parrotfishes (Scarinae and Sparisomatinae: 10 species), snappers (Lutjaninae: six species), and groupers (Serraninae and Epinephelinae: nine species) (Table S2). Trophic guilds were assigned following Sandin et al. (2008).

Fish and coral diversity

The total number of fish species (species richness) per reef was determined (i.e., if an individual was recorded on any one of the six transects per reef). Species richness was then used to calculate Shannon’s Diversity Index (H), which was in turn used to calculate species evenness using the equations: HR=−∑i=1SPixlnPi and ER=HR∕lnS, where H(R) is Shannon’s Diversity Index for a reef R, which has I → S number of species (thus, S is species richness), P is the proportion of species i (number of individuals of the species/total number of individuals of all species), and E(R) is species evenness for reef R (Heip, Herman & Soetaert, 1998). For scleractinian corals, genus richness, which has been shown to be an adequate surrogate for species richness (Balmford, Green & Murray, 1996; Bett & Narayanaswamy, 2014), was recorded on each reef. Species richness was not measured directly for the sake of convenience in the field and due to the high probability of identification errors encountered within many genera present in the Red Sea. Several recent studies in the region have revealed troublesome scleractinian groups and new taxonomic discoveries (e.g., Huang et al., 2014; Terraneo et al., 2014; Arrigoni et al., 2014; Bouwmeester et al., 2015), highlighting the need for caution when working at the species level in this region until coral taxonomy is formally revised.

Spatial trends and statistical analysis

In order to explore potential spatial variation in overall fish biomass, commercial fish biomass, and overall species richness, we used Kruskal–Wallis tests (KW) with post-hoc Mann–Whitney U tests (MW) to identify significant differences between the medians of the samples for all reefs. These non-parametric tests were chosen due to deviations from normality that occurred in the data collected from some of the reefs. However, for fish species richness, one-way ANOVA and post-hoc Tukey’s tests were used to compare the means of the reefs, since richness datasets successfully met assumptions of normality. Pearson’s correlation tests were used to explore potential correlations between distance from shore and overall biomass at both sampled depths as well as to explore correlations between coral cover or coral genus richness and fish biomass or fish species richness. SPSS Statistics®, version 21, was used to conduct these statistical analyses.

Fish and benthic assemblages

In order to identify and analyze patterns of similarity in assemblages across reefs, we created non-metric multidimensional scaling (NMDS) plots using fish biomass, fish densities, and benthic cover data. All data were log-transformed (Log(x + 1)) to eliminate biases caused by very highly abundant species, and the Bray-Curtis method was used to create all resemblance matrices. As per guidelines provided by Clarke (1993) for ecological data, we considered plots with 2D stress values higher than 0.2 to be poor representations of the data in 2-dimensional space, while stress values lower than 0.1 to be excellent representations. Most analyses were followed up by analyses of similarity (ANOSIM) to test for significant clustering and similarity percentage (SIMPER) analyses to identify the top species or categories contributing to dissimilarity between clusters (Clarke, 1993). The software PRIMER, version 6, was used for these analyses (Clarke & Gorley, 2006).

Results

Fish biomass increased moderately with distance from shore, and trophic composition was dominated by herbivores

A grand total of 13,792 fishes from 136 species and 44 families/sub-families (Table S1) were counted on the surveys. Overall, fish biomass was higher at 2 m depth than at 10 m on most reefs (Figs. 2 and 3; see Table S3 for a summary of exact values). However, mean fish biomass at 10 m significantly positively correlated with distance from shore (Pearson’s test, r = 0.881, R2 = 0.800, p = 0.002), while at 2 m it did not (Fig. 2B). The grand mean of fish biomass for all Thuwal reefs, with depths pooled, is 16.4 kg/100 m2.

Figure 2 Scatter plot of mean fish biomass at 10 m and 2 m depth against distance from shore.

Mean fish biomass in kg/100 m2 at (A) 10 m and (B) 2 m plotted against nearest straight line distance from shore as calculated in ArcMap. Error bars represent standard error. R2 values are shown on each panel and suggest strong correlation at 10 m and poor correlation at 2 m; (*) indicates significant regression (p = 0.009). Dotted vertical lines delimit inshore, midshelf, and offshore areas. Fish surveys were conducted in May 2013 on nine reefs in the central Saudi Arabian Red Sea, with 3 replicate 25 × 8 m belt transects at each depth per reef.

Figure 3 Stacked bar chart of mean biomass of four fish trophic groups.

Mean fish biomass in kg/100 m2 for each of the nine study reefs at (A) 10 m, and (B) 2 m depth, color-coded according to stacked trophic group as per the inset key (planktivores, herbivores, carnivores, or top predators). Error bars show standard errors (SE) of the means. Means and SE bars shown are not relevant to the non-parametric tests described in the main text, but are shown to facilitate comparisons with other studies. Reef name abbreviations are presented on the x-axis (see main text for full names) and separated according to distance from shore into offshore, midshelf, and inshore reefs (delimited by vertical dotted lines for easy visualization). All data were collected in May 2013 from the central Saudi Arabian Red Sea.

Biomass trophic composition on all reefs was dominated by herbivores at both depths with few to no top predators, with the exception of one offshore reef (NR), which was the only reef in which top predator biomass was dominant at 10 m (Fig. 3). This was due to the observation of two whitetip reef sharks (Triaenodon obesus) on one of the 10 m transects on that reef. No sharks were observed on any of the other reefs. Other observed fish that were considered top predators were grouper, snapper, eel, and jack species. The biomass of herbivores correlated significantly with distance from shore at 10 m (Pearson’s test, r = 0.811, R2 = 0.657, p = 0.008), but not at 2 m (Pearson’s test, r = 0.487, R2 = 0.238, p = 0.183). The grand mean biomass of trophic groups on Thuwal reefs is 1.0 ± 0.2, 10.8 ± 2.6, 2.5 ± 0.1, and 2.1 ± 1.2 kg/100 m2 for planktivores, herbivores, carnivores, and top predators, respectively.

Commercial fish biomass did not differ significantly between reefs

Offshore reefs collectively had the highest mean biomass of the three commercial fish groups surveyed (see Table S2 for a list of commercial fish species), while inshore reefs had the lowest (Fig. 4). However, these differences were not statistically significant. There were also no significant differences between individual reefs (KW, p > 0.05 for all comparisons). For a complete list of commercial fish species observed in each group and for biomass values on individual reefs, please refer to Tables S2 and S3.

Figure 4 Mean biomass of the three most targeted commercial fish groups in Saudi Arabia: parrotfish, snappers, and groupers, color-coded as indicated by the inset key and averaged across three offshore, three midshelf, and three inshore reefs in the central Saudi Arabian Red Sea.

Bars represent standard error (SE). Means and SE bars shown are not relevant to the non-parametric tests described in the main text, but are shown to facilitate comparisons with other studies.

Coral cover was significantly lower on inshore reefs, and algal cover decreased moderately with distance from shore

We recorded a total of 38 benthic categories, including 25 genera of scleractinian corals (listed in Table S4). Mean percent coral cover ranged from 8.35% (±3.3) on inshore reef ASR to 30.70% (±3.7) on midshelf reef TWR (Fig. 5; Table S4 for a summary of exact values). There was no strong correlation between coral cover and distance from shore (Pearson’s test, r = 0.470, R2 = 0.221, p = 0.202). However, one-way ANOVA tests showed significant difference between individual reefs (F = 16.7, p = 3 × 10−6), and post-hoc tests showed that coral cover on inshore reefs was significantly lower than that of midshelf reefs (pTukey = 2 × 10−5) and offshore reefs (pTukey = 7 × 10−6). Coral cover also did not correlate strongly with fish species richness or with fish biomass (Pearson’s tests, p = 0.738 and 0.714, respectively). As for mean algal cover, there was a moderate negative correlation with distance from shore (Pearson’s test, r =  − 0.658, R2 = 0.433, p = 0.054), which co-occurred with the aforementioned positive correlation of herbivorous fish biomass with distance from shore.

Figure 5 Stacked bar chart of mean percent benthic cover on the nine study reefs.

Mean percent cover (±SE) of benthic categories recorded on the nine study reefs in the central Saudi Arabian Red Sea. Reef names are shown as abbreviations on the x-axes and separated according to distance from shore. Data were collected on 10 m long transects at 10 m and 2 m depths using the line-intercept method. The category Hard corals summarizes values for 25 scleractinian coral genera that were observed (listed in Table S4); Soft corals summarize at least 6 genera; Hydrozoans contained only the genus Millepora; Zoanthids only contained the genus Palythoa, and the remaining categories are listed in the legend. CCA: crustose coralline algae.

Richness and diversity indices did not differ significantly between reefs

A total of 136 species of fish were recorded in our surveys (Table S1). Fish species richness ranged from 54 on one of the inshore reefs (ASR) to 70 species on one inshore reef (TR) and one midshelf reef (TWR). Species evenness, which was calculated from Shannon’s Index for each reef, ranged narrowly from 0.59 to 0.77, indicating a fairly even number of individuals per species on all reefs (Table 1). Species richness was highest on average on midshelf reefs, but no statistical significance was found (one-way ANOVA, F = 2.461, p = 0.166).

A midshelf reef (FR) had the highest number of hard coral genera (23), while an inshore reef (ASR) had the lowest (10 genera). MW tests showed inshore reef ASR to have significantly lower coral genus richness than all but two other reefs (pMW < 0.005). Coral genus richness also had a poor linear correlation with fish species richness (Pearson’s test, r = 0.224, R2 = 0.050, p = 0.562) and fish biomass (Pearson’s test, r = 0.096, R2 = 0.009, p = 0.806).

Fish and benthic assemblages on inshore reefs are clearly different from all other reefs

A number of iterations were attempted to identify any significant differences in fish and benthic assemblages between the reefs and between the two depths at which the data were collected. These analyses used fish biomass, fish densities, and benthic cover. Here, we present the most significant results, while a more complete list of NMDS, ANOSIM, and SIMPER analyses and their results can be found in supplementary material (Table S6).

Table 1 A summary of fish and hard coral diversity indices for each of the nine study reefs in the central Saudi Arabian Red Sea (see ‘Study Site’ for abbreviations).

For coral genus and fish species richness, the numbers shown are the maximum numbers of genera and species found on each reef, respectively. Fish species evenness was calculated from Shannon’s Diversity Index for each reef which was based on the reported species richness. Each reef was surveyed using six replicate visual belt transects. Habitat indicates the location of each reef on the continental shelf, Reef is the abbreviated name of each study reef.

Habitat	Reef	Hard coral genus richness	Fish species richness	Fish species evenness	
Offshore	RR	14	60	0.68	
NR	16	59	0.76	
AMR	20	55	0.59	
Midshelf	FR	23	69	0.62	
TWR	18	70	0.66	
AHR	20	64	0.61	
Inshore	ASR	10	54	0.77	
TR	18	70	0.72	
EFR	20	55	0.59	

The NMDS plots for mean fish biomass and densities at 10 m (Figs. 6A and 6C) are very similar to each other with a very clear and tight clustering of all reefs in one cluster except for two inshore reefs (ASR and EFR), which separated from the other reefs but did not cluster closely together. This shows very high similarity (ANOSIM R = 0.9, sig. 2.8) at 10 m depth in fish assemblages (by biomass as well as densities) among all reefs except ASR and EFR. In terms of biomass, Caesio lunaris (Family: Caesionidae) contributed the most to the dissimilarity (SIMPER dissimilarity contribution (hereafter Contrib.) = 7.9%), being more abundant in the group containing offshore reefs, midshelf reefs, and one of the inshore reefs (TR).

Figure 6 Fish and benthic assemblage NMDS plots.

Non-metric multi-dimensional scaling (NMDS) plots from Bray-Curtis resemblance matrices based on log (x + 1)-transformed reef averages for fish biomass (A and B), fish densities (C and D), and benthic cover (E and F); the left column of panels shows 10 m assemblages, while the right column shows 2 m assemblages. X-axes represent NMDS1 and y-axes represent NMDS2. Reef name abbreviations are shown in the inset key next to their representative symbols, and the nine reefs are color-coded according to distance from shore as shown by the key. The 2D stress values are shown in each plot. All data were averaged across the relevant replicates for each reef. Fish and benthic data were collected together on the same transects (six per reef in total) from the central Saudi Arabian Red Sea.

However, looking at fish assemblages at 2 m (Figs. 6B and 6D), we find all inshore reefs separating (including TR) from all other reefs, which clustered together (ANOSIM sig. 1.2% for both biomass and densities). However, the offshore cluster was less tight than it was at 10 m, indicating more dissimilarity within the shallow fish communities. The farming Stegastes nigricans contributed highly to the dissimilarity between inshore and offshore communities in terms of both biomass and numerical density (Contrib. 14.6 and 6.8%, respectively), being abundant on inshore reefs and nearly absent on other reefs. Another damselfish, Chromis dimidiata, also contributed by being more abundant on midshelf and offshore reefs (Contrib. 4.3%).

As for benthic assemblages at 10 m (Fig. 6E), inshore reefs in addition to two midshelf reefs (TWR and FR) separated from the remaining four reefs (ANOSIM R = 0.78, sig. 0.8%), with sand and rubble (collective Contrib. 32.0%) and CCA (9.2%) contributing the most to the separation. Sand and rubble were more abundant in the group containing the inshore reefs, while CCA was more abundant in the group containing the offshore reefs. However, at 2 m (Fig. 6F), there was a clearer separation again between inshore reefs and all other reefs (ANOSIM R = 0.82, sig. 1.8%). Turf algae (Contrib. 14.6%), rock (10.4%), and Porites (10.0%) were more abundant on inshore reefs, contributing highly to the dissimilarity, while Pocillopora (14.3%), CCA (13.5%), and xeniid soft corals (9.6%) were more abundant on offshore and midshelf reefs (Table S6).

Discussion

We present here a description of the spatial variation in fish and benthic communities on a group of central Saudi Arabian Red Sea reefs, with particular focus on differences in assemblages along a cross-shelf gradient. Our results show that fish biomass increases moderately with distance from shore in Thuwal and that fish communities are dominated by herbivorous fishes at all sites. Benthic communities are fairly similar with the exception of shallow inshore sites. As very little detailed ecological data is available for this region, this study will be valuable for future Red Sea reef research and for conservation planning efforts.

Low fish biomass, absence of top predators, and low abundances of commercial fishes suggest possible overfishing and low resilience

It is likely that the increase in mean fish biomass with distance from shore, which was mostly evident at 10 m depth, is mostly due to the change in surrounding water depth (Fig. 1). Inshore reefs slope to a sandy bottom at much shallower depths (between 12 and 20 m) than the offshore reefs (Fig. 1), and so offshore reefs may simply be able to support the occurrence of higher biomass than inshore reefs. In another part of the Red Sea, McMahon et al. (2016) found that offshore reefs had elevated contributions of planktonic carbon to their food webs compared to inshore reefs. It is possible this is further translated into increased productivity and fish biomass offshore. Our results showed that this increase in biomass could not be explained by coral cover or coral diversity.

Moreover, as a general trend, fish biomass on Thuwal reefs appears to be relatively low with particularly low proportions of top predators. Compared to relatively remote and nominally pristine locations around the world, including sites in the central Pacific (Sandin et al., 2008; Williams et al., 2011; Friedlander et al., 2014), the North-Western Hawaiian Islands (Friedlander & DeMartini, 2002; Williams et al., 2011), and even some relatively remote and unfished parts of the Red Sea (Kattan, 2014; Spaet, Nanninga & Berumen, 2016), Thuwal reefs had very low fish biomass. Only Nazar Reef (NR) and Abu Madafi Reef (AMR), which had the highest mean biomass values in this study (Fig. 3), had values comparable to, and sometimes higher than these sites (Table 2). However, even when mean biomass on NR exceeded that of other sites, it is important to note that the percentage of top predators in all other sites far exceeded NR’s 39% (most of which was contributed by two whitetip reef sharks—the only sharks observed in our study). Herbivores make up the bulk of the biomass on Thuwal reefs (Fig. 3).

Table 2 A comparison of mean fish biomass and top predator composition between Thuwal reefs and reefs considered pristine in studies in other regions.

The comparison includes the mean biomass on Nazar reef as the reef with the highest mean biomass in this study as well as the overall mean biomass of all nine Thuwal reefs. Biomass indicates the mean fish biomass (standardized to kg/100 m2) from each site, while top predator composition indicates the percentage of top predator biomass compared to total fish biomass. There were no major differences in the way in which top predators were defined across the studies.

Site	Region	Mean fish biomass (kg/100 m2)	Top predator composition (%)	Study	
Kingman reef	Pacific	53	81	Sandin et al. (2008)	
Pearl & Hermes Atoll	North-Western Hawaiian Islands	47	81	Friedlander & DeMartini (2002)	
Kure Atoll	North-Western Hawaiian Islands	35	66	Williams et al. (2011)	
Jarvis reef	Pacific	25	68	Williams et al. (2011)	
French Frigate Shoals	North-Western Hawaiian Islands	26	62	Friedlander & DeMartini (2002)	
Palmyra Atoll	Pacific	25	64	Sandin et al. (2008)	
Ducie Island	Pacific	16	63	Friedlander et al. (2014)	
Deep South	Red Sea, Sudan	43	67	Kattan (2014)	
Nazar reef	Red Sea, Saudi Arabia	31	39	This study	
All Thuwal reefs	Red Sea, Saudi Arabia	16	13	This study	

Our survey design (using short belt transects and having a small number of replicates) might not be adequate for capturing the abundances of large mobile predators such as sharks, trevallies, and barracudas, which are typically surveyed using other techniques such as baited cameras (e.g., Robbins et al., 2006; Goetze & Fullwood, 2013; Spaet, Nanninga & Berumen, 2016). Nevertheless, we used the same method that was applied by Kattan (2014) in the Sudanese Red Sea, where much higher abundances of top predators were neverthelesss recorded. Some of the other studies listed in Table 2 also used a similar transect length (e.g., Sandin et al., 2008) and captured much higher abundances of top predators. Therefore, we believe that the absence of top predators in our study reflects true low abundances.

Top predators such as sharks, jacks, and groupers are critical in maintaining the structure of reef communities, and overfishing these groups can lead to trophic cascades and overall loss of diversity and resilience (Friedlander & DeMartini, 2002; Sandin et al., 2008; Salomon et al., 2010; Houk & Musburger, 2013). Thus, the trophic structure on Thuwal reefs raises concerns about resilience, which should be addressed in future studies, and points to a possible overfishing problem. Currently, there is substantial and growing evidence of severe overfishing in the Saudi Arabian Red Sea. Decades of catch records suggest that Saudi Arabian fisheries may have been operating beyond sustainable levels since the 1990s (Jin et al., 2012). Similarly, Spaet & Berumen (2015) have shown evidence of unsustainable elasmobranch fisheries based on two years of fish market surveys. The trophic structure observed on Thuwal reefs in our study, therefore, could be a result of overfishing. Additionally, the absence of one of the major commercial species from our surveys, the grouper Plectropomus pessuliferus (Family: Serranidae), and the lack of significant variability in the biomass of the three commercial fish groups shown in Fig. 4 also suggest heavy and spatially homogeneous fishing pressure. There is substantial evidence in the literature showing that protection from fishing can lead to relatively rapid recovery of stocks and to generally healthier reefs (e.g., Gell & Roberts, 2003; Bruce et al., 2012; Almany Glenn et al., 2013), and so, prompt action to stop overfishing in Saudi Arabia may lead to positive results.

Herbivores are also an essential functional group for maintaining the resilience of reefs, as they assist coral recruitment and recovery from disturbances by keeping macroalgae under control (Williams & Polunin, 2001; Hughes et al., 2007; Ledlie et al., 2007). We speculate that the increase in herbivore biomass with increasing distance from shore in Thuwal indicates that offshore reefs may be relatively more resilient than inshore reefs. However, it is unknown whether the resilience of even the offshore reefs is adequate in the face of continuing overfishing.

Variations in coral and algal cover show impacts of a bleaching event and indicate degradation of inshore habitat

Coral cover differed significantly between inshore reefs as a group and other reefs, which is likely partially due to the impact of the 2010 bleaching event (Furby, Bouwmeester & Berumen, 2013; Pineda et al., 2013). It appears that these inshore reefs have not yet recovered their live coral cover in the ∼3 years that passed between the bleaching event and the commencement of data collection for this study. (This situation is likely to continue as inshore reefs were again impacted by a similar bleaching event in 2015 (Lozano-Cortés et al., 2016)). Studies from other locations, such as the Great Barrier Reef, have similarly found that coral cover on inshore reefs tended to decline more severely than on offshore reefs following disturbances (e.g., Sweatman et al., 2007). However, recovery time was found to be highly variable; while some studies reported relatively rapid recovery of coral cover (e.g., ∼2.5 years reported by Hughes et al. (2007)), others reported that, even after six years, inshore reefs hardly recovered any lost coral cover (Sweatman, Delean & Syms, 2011).

At the same time, inshore reefs in this study have higher coverage of turf algae and high abundance of the damselfish species that farms it, Stegastes nigricans (Family: Pomacentridae), accompanied by generally lower herbivore biomass and diversity. The presence of S. nigricans and their turf algae farms has previously been found to be associated with high rates of coral mortality and high abundances of coral disease-associated pathogens, and they are often considered indicators of a degraded habitat (Casey et al., 2014; Casey, Connolly & Ainsworth, 2015; White & O’Donnell, 2010). This potentially raises further concerns about the health of inshore reefs in Thuwal and their vulnerability to future disturbances. Larger datasets and continuous monitoring of the reefs would allow stronger inferences about future reef resilience in Thuwal (e.g., Bellwood et al., 2004; Pratchett et al., 2011).

Correlations between benthic and fish diversity indices may not be observable due to compounding effects of stressors or to small sample size

Although previous studies have found benthic cover, diversity, and complexity to be correlated with fish species richness (e.g., Roberts & Ormond, 1987; Chabanet et al., 1997; Chong-Seng et al., 2012), we found no such patterns on Thuwal reefs neither with coral cover nor with coral genus richness. This could be due to different stresses impacting reef communities simultaneously. For instance, if fishing pressure is impacting fish communities (and indirectly benthic communities) and the bleaching event has impacted the benthic communities (and indirectly the fish communities), then any naturally occurring correlations between benthic cover or diversity and fish diversity could become unobservable. Alternatively, the lack of correlation between benthic and fish diversity could be due to the number of replicates in the survey design, which may not be adequate for investigating such correlations. Although traditional survey methods such as LIT are commonly used to assess general coral cover, they have been shown to be less effective in assessing coral richness or diversity unless sampling effort is highly intensified (Leujak & Ormond, 2007; Roberts et al., 2016a; Roberts et al., 2016b). Finally, the relationship between benthic and fish diversity in Thuwal may be non-linear, thus requiring further analysis beyond linear correlation. Therefore, in this study, the relationship between fish and benthic diversity may be observable only in terms of qualitative assemblage rather than quantitative indices.

Inshore fish and benthic assemblages are markedly different from offshore assemblages partially due to the impacts of coral bleaching

It appears that, especially in the shallower depths, inshore reefs are markedly different in fish and benthic assemblage from other reefs in the area. Furby, Bouwmeester & Berumen (2013) had found that, prior to bleaching, coral assemblages (genus-level abundances and coral cover) were similar on inshore and offshore Thuwal reefs, and that the post-bleaching differences were mostly caused by a decline in acroporids and pocilloporids on inshore reefs, which are faster-growing corals that tend to be more susceptible to bleaching (Marshall & Baird, 2000). Very similar trends were also reported in other locations, for example in French Polynesia by Berumen & Pratchett (2006). Our study supports these findings and also shows turf algae to be one of the main contributors to the dissimilarity between inshore and offshore shallow communities. Similarly, we also found the slow-growing genus Porites to be a more characteristic community component on inshore reefs, while Acropora, Pocillopora, and Stylophora were important components distinguishing assemblages only on midshelf and offshore shallow communities.

Herbivore assemblages are commonly recognized as a key functional component of coral reef communities (Lewis, 1986; Hughes et al., 2007; Adam et al., 2011). On Thuwal reefs, we found very similar herbivore assemblages on all reefs except the inshore reefs. Offshore communities were characterized by the surgeonfishes Acanthurus sohal, Naso unicornis, Ctenochaetus striatus, and A. nigrofuscus, while inshore communities were dominated mostly by the farming damselfish Stegastes nigricans. This coincides with the higher abundance of turf algae inshore and presents a potential difficulty for the recovery process of inshore reefs. These territorial damselfish promote the mono-cultural growth of algae on reef flats and crests, subsequently preventing settlement by corals and other invertebrates (e.g., White & O’Donnell, 2010), whereas other types of grazers, such as surgeonfishes and parrotfishes, tend to remove algae and promote invertebrate settlement (Vine, 1974; Jones, Santana & McCook, 2006). Thus, the low herbivore diversity on inshore reefs indicates potentially reduced resilience.

On our deeper transects, fish assemblages were very similar across all reefs except for two of the inshore reefs. With regards to both biomass and numerical density (Fig. 6), the offshore communities seem to be dominated by planktivorous fishes, such as Caesio lunaris, Chromis dimidiata, Chromis flavaxilla, and Pseudanthias squamipinnis; these contributed the most to the similarity within the offshore reef cluster. We speculate that this trend may be due to a higher influx of zooplankton on more exposed reefs (e.g., Hamner et al., 1988).

It seems that, apart from the known bleaching event that altered inshore communities in 2010, Thuwal reef communities are fairly similar in structure. Repeating this study in an area of the Saudi Arabian Red Sea where bleaching did not occur may confirm or deny that the bleaching was the main driver of the differences we observed among reef communities. In the meantime, however, bleaching impact seems to be a highly likely variable. Inshore reefs are most likely vulnerable to thermal bleaching as a result of anomalous reductions in wind-driven circulation (Furby, Bouwmeester & Berumen, 2013); the seascape arrangement may thus be among the most significant environmental drivers in this area. Our results suggest that inshore reefs may be generally less healthy and more vulnerable to disturbances than offshore reefs.

Conclusions

We presented a description of the spatial variation of fish biomass and fish and benthic communities on Thuwal reefs in the central Saudi Arabian Red Sea. Our findings can be summarized as follows:

1. Offshore Thuwal reefs seem to support higher fish biomass than inshore reefs. But fish biomass in Thuwal in general is quite low compared to other reef systems around the world and in the Red Sea that are considered “healthy” or relatively pristine.

2. Trophic structure on all Thuwal reefs is bottom-heavy with most biomass attributed to herbivores; top-predators are few or nearly absent.

3. Commercially valuable fish are in very low abundance throughout the area.

4. There are a few dissimilarities in benthic and fish assemblages which are mostly found between inshore reefs as a group and all other reefs:

(a) Inshore benthic communities are characterized by having more turf algae and slow-growing corals compared to offshore reefs.

(b) Farming damselfish dominate shallow inshore herbivore communities compared to more diverse herbivore communities on offshore reefs.

In addition to identifying inshore areas in Thuwal to be more vulnerable to disturbances such as thermal stress, our findings support existing evidence that Thuwal reefs may be heavily overfished, as indicated by trophic structure and low biomass (‘Low fish biomass, absence of top predators, and low abundances of commercial 315 fishes suggest possible overfishing and low resilience’ and ‘Variations in coral and algal cover show impacts of a bleaching event and 379 indicate degradation of inshore habitat’). Intense fishing affects communities on many levels, from species’ life-history traits to population fitness and community structure, in ways that generally lower diversity and reef resilience (Robertson et al., 2005; Salomon et al., 2010). The status of both sharks (Spaet, Thorrold & Berumen, 2012; Spaet & Berumen, 2015) and groupers (DesRosiers, 2011) are particularly alarming in the Saudi Arabian Red Sea, even compared to other parts of the Red Sea (Kattan, 2014; Spaet, Nanninga & Berumen, 2016), and fishing regulations as well as other forms of protection may be urgently needed to halt the collapse of fisheries.

Collectively, these findings provide preliminary scientific basis for future spatial prioritization and conservation planning in Thuwal. Our results also identify individual species that characterize inshore and offshore reefs, and these species, together with a selection of commercial species, could now be used to set quantitative conservation objectives (e.g., Schmiing et al., 2014). The results presented here thus enable future resource management, conservation efforts, and spatial planning to be based on data, as opposed to convenience or “best-guesses” of appropriate actions.

Supplemental Information

Supplemental Information 1 Raw benthic cover data

Click here for additional data file.

Supplemental Information 2 Raw fish data

Click here for additional data file.

Table S1 List of surveyed fish species, their assigned trophic groups, and “a” and “b” values for biomass calculations.

List of all fish species found on the 9 study reefs, the families or sub-families they belong to, and the trophic groups to which they were assigned, with corresponding a and b values obtained from FishBase (2014) and used in biomass calculations. All data were collected in May 2013 in the central Saudi Arabian Red Sea, with six 25 × 8 m belt transect replicates on each reef: 3 at 10 meters’ depth, and 3 at 2 meters’ depth.

Click here for additional data file.

Table S2 List of commercial fish species observed and the families/subfamilies they belong to.

A list of fish species that were observed during our surveys and that belong to 3 of the most commercially valued and heavily-targeted fishes by fishermen in Saudi Arabia (parrotfish, snappers, and groupers), and the families/subfamilies to which they belong.

Click here for additional data file.

Table S3 Summarized fish biomass data.

Mean and total fish biomass on the 9 study reefs in the central Saudi Arabian Red Sea expressed in mean kg/100 m2 (±SE). Each reef was surveyed using six replicate visual belt transects, three at 10 m and three at 2 m depth. “Habitat” indicates the location of each reef according to distance from shore, “Reef” is the name of each study reef (see main text for abbreviations). Values are divided by trophic group (planktivores, herbivores, carnivores, and top predators) and also shown as a Total for all groups combined.

Click here for additional data file.

Table S4 List of benthic taxa observed on study reefs.

List of benthic categories recorded on the 9 study reefs. The taxonomic sub-categories (mostly genera) are listed in the second column as recorded during the surveys. A total of 25 scleractinian genera were recorded, at least 5 soft coral genera (the sub-category “Xeniidae” may have included more than one genus which were unidentifiable in the field), one zoanthid genus, and one genus of hydrozoan. Sponges and algae were recorded as general categories and the rest are non-living substrate categories. All data were collected using the line-intercept method in May 2013 in the central Saudi Arabian Red Sea on shorter (10 m long) subsets of the same transects used to collect fish data. There were 3 replicates at 10 m, and 3 at 2 m.

Click here for additional data file.

Table S5 Summary of observed benthic cover.

Mean percent cover (±SE) of benthic categories recorded on the 9 study reefs in the central Saudi Arabian Red Sea. Reef names are shown as abbreviations in column headers and separated according to distance from shore. Data were collected on 10 m long transects at 10 m and 2 m depths using the line-intercept method. The category “hard corals” summarizes values for 25 scleractinian coral genera that were observed (listed in Table S4; “soft corals” summarize at least 6 genera; “hydrozoans” contained only the genus Millepora; “zoanthids” contained only the genus Palythoa; and the remaining categories were recorded as shown in the table. “CCA” stands for crustose coralline algae.

Click here for additional data file.

Table S6 A list of all non-metric multidimensional scaling (NMDS), analyses of similarity (ANOSIM), and similarity percentage (SIMPER) analyses performed on fish and benthic data.

A list of all non-metric multidimensional scaling (NMDS), analyses of similarity (ANOSIM), and similarity percentage (SIMPER) analyses performed on the fish and benthic data collected on the 9 study reefs in the central Saudi Arabian Red Sea. The first column indicates which dataset was used (all data were log (x+1)-transformed); the second column indicates which subset of the dataset was included in the analysis (whether all replicates were used, replicates from a certain depth were used, and/or reef averages were used); the third column contains brief descriptions of NMDS plots produced; the fourth column shows the 2D stress for the plots described in the third column (>0.2 indicates poor representation of data in 2-dimensional space, <0.1 indicates excellent representation); the fifth column shows the global R statistic for ANOSIM (as the value approaches 1, the separation of groups analyzed is larger); the sixth column shows the significance of the separation indicated by the R statistic (values below 5% are considered significant); the seventh column shows the SIMPER percent average dissimilarity between the groups analyzed; and the last column lists some of the highest-contributing species/categories to the dissimilarity and the percent of their contribution between parentheses. Cells that are filled with dashes indicate that ANOSIM and SIMPER were not performed for the corresponding analyses indicated in the second column.

Click here for additional data file.

Fieldwork was facilitated by the KAUST Coastal and Marine Research Core Lab, and Tane Sinclair-Taylor provided assistance with creating one of the figures.

Additional Information and Declarations

Competing Interests

Author Contributions

Data Availability

The authors declare there are no competing interests.

Maha T. Khalil conceived and designed the experiments, performed the experiments, analyzed the data, wrote the paper, prepared figures and/or tables.

Jessica Bouwmeester performed the experiments, reviewed drafts of the paper.

Michael L. Berumen conceived and designed the experiments, performed the experiments, contributed reagents/materials/analysis tools, reviewed drafts of the paper.

The following information was supplied regarding data availability:

The raw data has been supplied as a Supplementary File.

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
