# Peer review of "Spatial variation in coral reef fish and benthic communities in the central Saudi Arabian Red Sea"

_PeerJ, doi:10.7717/peerj.3410_

## Round 0.1 · original submission · Major Revisions

Dear Dr. Maha

Please consider all the referees remarks in your revised manuscript and indicate in your rebuttal letter a point-by-point response to the referees. Also make sure to compare your results with other studies in other reef areas (e.g. PMID: 22679480, PMID: 18301735).

·

Basic reporting

The manuscript report on spatial patterns of reef fish and benthic assemblages for a poorly studied region. Although results are relevant, several improvements in data analyses, language and format are needed. Detailed comments are given in the attached document (reviewed ms with track changes)

Experimental design

Needs some clarification.

Validity of the findings

Not able to evaluate, as analyses are not appropriate.

Additional comments

The manuscript report on spatial patterns of reef fish and benthic assemblages for a poorly studied region. Although results are relevant, several improvements in data analyses, language and format are needed. Detailed comments are given in the attached document (reviewed ms with track changes)

Reviewer 2 ·

Basic reporting

The article was written in English using clear and unambiguous text.
The article does not include sufficient introduction and background to demonstrate how the work fits into the broader field of knowledge. Relevant prior literature should be appropriately referenced.
All appropriate raw data should be available in a public repository, please choose one and provide accession numbers/links.

Experimental design

The submission does not clearly define the research question, which must be relevant and meaningful. The knowledge gap being investigated was not clearly identified, and statements should be made as to how the study contributes to filling that gap.
The investigation seems to be conducted rigorously and to a high technical standard.
Methods were described with sufficient information to be reproducible by another investigator. However the study questions/hypothesis should be raised in introduction section and revisited (how and why they were tested) in methods section.

Validity of the findings

The data are robust, statistically sound, and controlled, besides low sample number.
The data on which the conclusions are based must be provided or made available in an acceptable discipline-specific repository.
The conclusions should be appropriately stated, should be connected to the original question investigated, and should be limited to those supported by the results. I recommend re-writing conclusion section following my comments bellow.

Additional comments

The manuscript entitled “Spatial variation in coral reef fish and benthic communities in the central Saudi Arabian Red Sea” by Khalil et al have accessed the benthic and ichthyologic communities from Thuwal area. The manuscript is relevant and was well written however the manuscript is presented in as a purely description of the local/regional reefs. Besides low sample numbers, the present data is robust. Authors can improve the manuscript greatly changing their focus on broader ecological processes. With this being said I would suggest some major modifications before the manuscript publication. I hope authors find my comments useful to improve their manuscript.

How authors can shift from a local/regional focus to a more broad approach? Authors emphasize the limited knowledge of Red Sea and Saudi Arabian coral reefs instead of focusing on dominant processes occurring in coral reefs such as phase-shifts. If authors change the study focus, the scientific contribution would be much greater and a more broad audience would be interested the manuscript.

Authors should provide a link or accession number where data can be downloaded. Please choose one public repository and make data available.

Introduction:
Introduction is very focused on local/regional issues but authors’ objectives would be broader exploring the cross-shore gradient as hypotheses. Authors should contextualize their questions/hypothesis on a broader knowledge scope. Please re-write the last two paragraphs of introduction shifting the order of the presented information. Authors had used hypothesis tests (i.e. ANOVA and Regression)

Authors sampled two depths and have not mentioned anything about this in introduction. What motivated authors to adopt this sampling designed? Please add in introduction the motivation and detail better in methods.

Lines 36-37: “yet there are relatively few publications available from this region” in a quick search in Google Scholar I found more than 15K articles using “Saudi Arabia reefs” searching terms. Please make this statement carefully and more accurately. In what aspects Saudi Arabian coral reefs are understudied compared with the highly studied aspects?

Lines73-75: Are this information really relevant to be included? I suggest removing this. The manuscript needs to be shortened.


Methods:

Line 91: Please mention in what part of the reef the 10m sampling occurred, in reef walls?

Lines 156-157: Please explain what was tested with ANOVA.

Line 159: fish species richness? Please make it clear.


Abstract:
Abstract section does not provide contextualization on how the present study fits into the broader field of knowledge. Study’s objectives are not clear in Abstract section.

Line 17: Please find a substitute to “untouched”, something like “low impacted”, “with low human impact”, “with low human influence”.


Results:

Please change subtopics titles to full sentences summarizing the presented content. This modification will improve readability.

I suggest including short sentences explaining what were the most abundant fish families in results subsections. For instance, what were the dominant herbivorous species/families? In offshore and inshore reefs the same family/specie was dominant?

I suggest changing focus from particular reefs (reef names/abbreviations are hard to memorize/understand) to reef location (i.e. “offshore”, “midshelf” and “inshore”). The nMDS supports such results presentation, so please consider changing it. This would improve manuscript readability.

Line 175: fishes?

Line 178: Please include inside brackets what was the statistical test.

Lines 187-188: Please include inside brackets what was the statistical test.

Line 194: Remove “to the nearest kilometer” and “of the mean”

Line 211: Please make sure to point Table S2 for more details on what were the commercial reef fish species.

Line 286: Please include the family name for “Caesio lunaris”

Discussion:

Authors should change discussion focus. In the present form the manuscript is purely descriptive however authors have data to present and discuss their results more deeply based on broader hypothesis/questions. Authors should consider re-writing discussion (and introduction) to shift local/regional focus to general coral reefs issues. Authors can explore their local scale data do discuss, for instance, the effect of human influence on coral reefs (there are a plenty of literature regarding this issue).

This section is very long; authors should consider reducing it focusing on the main hypothesis/questions.

Please change subtopics titles to full sentences summarizing the presented content. This modification will improve readability.

Lines 339-351: Discussing the study limitations is really important, especially the low number of replicates, however I suggest summarizing this methodological discussion to shorter sentences and focus more on the problems of the reduced top predators biomass/abundance (what authors did in the same paragraph).

Lines 362-375: Coral reefs’ resilience has been debated extensively in literature.
Resilience has a complex definition and authors does not contextualize it well both in introduction and in discussion section. Authors should consider explaining it better (both in introduction and discussion) or remove this information.

Lines 376-399: This subtopic should be shortened and summarized in just one paragraph. Speculations about coral reefs’ resilience were made without proper introductory background. As far I understood authors data does not allow so many speculation regarding resilience (only one temporal sampling). I suggest focusing on the second paragraph subject. Authors would explore better the possible effects of territorial damselfishes abundance at inshore reefs, there is an interesting link between them and coral pathogens (see Casey et al DOI 10.1038/srep11903 for example).



Conclusion:

Lines 467-468: There are previous data to confirm this statement? The word in the second sentence is “others”? “Others” refer to Midshelf and Offshore reefs or to other reefs from different locations? Please make it clear. I am also concerned about this statement regarding to geographical/local peculiarities. Maybe this separation is resulted to a different environmental set (e.g. depth), not due to coral bleaching. Be cautious about this conclusion it does not sounds very accurate.

Lines 469-474: Please move this information to Discussion section.

Line 477: Please replace “analysis” to “results”.
Figures and tables:

Figure 1 – I suggest including a small map on regional scale showing here study area is located both in Saudi Arabian Red Sea and in the world. With this minor modification authors should remove unnecessary information in legend (e.g. “near the town of Thuwal, the King Abdullah University of Science and Technology (KAUST), and the King Abdullah Economic City (KAEC)”)

Figure 2 – Why the model line is limited to the first 10 km? I recommend inverting the plots order; it is more intuitive to have 2m first (above) than 10m (below). I also recommend including the classification (“offshore”, “midshelf” and “inshore”) on these plots. Suggestion: below x-axis authors can include lines discriminating it. What is the R2 of Figure 2b? Please include it.

Figure 3 – Please remove the line breaks from the words “offshore”, “midshelf” and “inshore”. I also recommend including a line to delimit the sites position, this would facilitate considerably readers’ visualization and understanding.

Figure 4 – Please see Figure 3 comments.

Figure 5 – I recommend changing red and green colors once colorblind people will not be able to distinguish these colors. I suggest writing “fish biomass”, “fish densities” and so on inside each plot. This would improve readers understanding. Please write do not abbreviate “nMDS” in the first phrase of figure legend.

Table 1 – I suggest moving it to Supplementary information. Information within this table is redundant with Figure 2 and 3.

Table 3 – What about hard coral evenness index? I recommend presenting some diversity index (e.g. Shannon) in this table.

Table 4 – I recommend removing the line separating the last three studies from the previous.

---

## Round 0.2 · Minor Revisions

Dear Authors

Please pay attention to the final Referee #1 remarks and correct your paper accordingly. Sincerely . Fabiano

Reviewer 2 ·

Basic reporting

After significant efforts authors had improved the manuscript substantially. I still think authors could reduce the "descriptive approach" but this would not be a rejecting criteria as far as I understood. Now I have now only minor suggestions before manuscript acceptance.

Experimental design

Linear regression aims to model the relationship between a dependent variable with explanatory (independent) variables not correlate variables. Please consider modifying material and methods text:

Lines 234-238:
Simple linear regression tests were also used to examine the correlation between distance from shore and overall biomass at both sampled depths. Regression was also used to explore correlations between coral cover or coral genus richness and fish biomass or fish species richness. SPSS Statistics®, version 21, was used to conduct these statistical analyses.

Please consider doing correlation analysis or changing the text explaining what hypothesis were tested using linear regressions.

Validity of the findings

I suggest moving all the speculative conclusions to discussion once PeerJ guild lines suggests:

"Conclusion are well stated, linked to original research question & limited to supporting results.
The conclusions should be appropriately stated, should be connected to the original question investigated, and should be limited to those supported by the results."

---

## Round 0.3 · accepted · Accept

Thank you for the revisions. The paper can now be accepted.